# OPENMU: YOUR SWISS ARMY KNIFE FOR MUSIC UNDERSTANDING

## ABSTRACT

We present OpenMU-Bench, a large-scale benchmark suite for addressing the data scarcity issue in training multimodal language models to understand music. To construct OpenMU-Bench, we leveraged existing datasets and bootstrapped new annotations. OpenMU-Bench also broadens the scope of music understanding by including lyrics understanding and music tool usage. Using OpenMU-Bench, we trained our music understanding model, OpenMU, with extensive ablations, demonstrating that OpenMU outperforms baseline models such as MU-LLaMA. Both OpenMU and OpenMU-Bench are open-sourced to facilitate future research in music understanding and to enhance creative music production efficiency[1].

## 1 INTRODUCTION

Multimodal Large Language Models (**MLLMs**) have successfully extended large language models (**LLMs**) by enabling them to perceive, process, and understand data in modalities beyond text (Tsimpoukelli et al., 2021; Liu et al., 2023b; Zhu et al., 2023; McKinzie et al., 2024; Zhang et al., 2023; Gong et al., 2024), such as images, videos, and audio. However, there has been limited effort (Gardner et al., 2024) focused on constructing MLLMs capable of effectively understanding the music modality or addressing Music Information Retrieval (**MIR**) tasks. MIR is a research field focusing on modeling, understanding and interpreting data relevant to music, aiming to improve the efficiency of music production (Serra et al., 2013). Conventional machine learning algorithms (Wang, 2003; Casey et al., 2008) sparked the success in music searching. Subsequently, deep learning models expanded the success to music tagging (Won et al., 2020), transcription (Gardner et al., 2021; Toyama et al., 2023) and representation learning (Castellon et al., 2021; Li et al., 2024; Won et al., 2024).

We aim to contribute to the MIR field by training an MLLM, dubbed **OpenMU**, for understanding music clips. Building on the versatile capabilities of LLMs and pretrained audio encoders, OpenMU effectively comprehends and reasons about input music clips, producing relevant answers accordingly. We also enable OpenMU to leverage well-established music tools to encourage synergies between OpenMU and creative practitioners through cooperation. OpenMU is expected to greatly improve music production efficiency. Creative practitioners can instruct OpenMU to describe a music clip's contents and features, saving minutes of time compared to listening to the full track.

The major obstacle we faced when training and evaluating OpenMU was the issue of data scarcity in the music modality (Serra et al., 2013; Seeger, 2003; Holzapfel et al., 2018). To address this issue, we construct **OpenMU-Bench**, a large-scale benchmark for *training and evaluating* MLLMs in music understanding. To construct OpenMU-Bench, we bootstrap new datasets using GPT-3.5, and leverage existing datasets when available. As a result, OpenMU-Bench comprises approximately one million training examples, covering various aspects of music understanding, such as music captioning, reasoning, multi-choice question answering, lyrics understanding and music tool using. To the best of our knowledge, no large-scale open-sourced benchmark comparable to OpenMU-Bench currently exists, and we hope it will advance future research and development of MIR.

In summary, our contributions include: **(1)** Proposing OpenMU for music understanding. OpenMU is an MLLM dedicated to the music modality, outperforming baseline models such as MU-LLaMA (Liu et al., 2024) in tasks like music captioning, reasoning, and multiple-choice question answering. We carefully evaluate various design choices for OpenMU and provide extensive ablations

---

[1]We will release the code, datasets, and model checkpoints upon acceptance.

on key factors. **(2)** Constructing a large-scale benchmark suite, OpenMU-Bench, consisting of approximately one million music understanding data examples. We bootstrap new data from GPT-3.5 for rich annotations and also leverage existing datasets. **(3)** Open-sourcing OpenMU and OpenMU-Bench. We hope that they will benefit future research and development in music understanding and enhance creative music production by providing rich resources and consistent evaluations.

## 2 RELATED WORK

**Understanding music** goes beyond recognizing objective attributes of music such as tempo (Böck et al., 2015; Sun et al., 2021) or instrumentation (Gururani et al., 2019; Zhong et al., 2023). It is also subjective and highly context-dependent, like determining music genres (Kereliuk et al., 2015) or moods (Bogdanov et al., 2019; Koutini et al., 2019). Researchers succeeded in understanding music by classifying music clips into predefined tags (Li et al., 2024; Won et al., 2024). Recently, *music captioning* (Manco et al., 2021) and *reasoning* (Gardner et al., 2024) tasks, where natural language descriptions are employed to describe music clips, have earned increasing attention. Also, the ability of selecting correct answers in *multi-choice question answering* is included in music understanding Weck et al. (2024). However, there has been limited exploration into enabling MLLMs to utilize external digital tools (i.e., established music tools) for music analysis. We hypothesize that a music understanding model can further boost the workflow of creative practitioners by deeply integrating the set of widely adopted music tools. Last but not least, lyrics information processing (Watanabe & Goto, 2020), such as semantic lyrics understanding (Zhang et al., 2022) enhances the understanding of a music clip. Therefore, we integrate it in OpenMU-Bench. Overall, we *broaden the scope of music understanding* by considering two extra aspects beyond music captioning and reasoning: *Music tool using* and *lyrics understanding*.

**Foundation models for music understanding.** Multimodal LLMs (**MLLMs**) (Tsimpoukelli et al., 2021; Liu et al., 2023b; Zhu et al., 2023; McKinzie et al., 2024; Gong et al., 2024) fuse non-textual information into LLMs (Liang et al., 2022) to solve real-world tasks requiring the ability of perceiving data in different modalities. The scope of MLLMs is recently expanded to include music. MU-LLaMA (Liu et al., 2024) and MusiLingo (Deng et al., 2024) narrowed down their scope to music captioning and question answering (QA); other critical aspects of music understanding, e.g., key and chord recognition, are not covered. Perhaps the closest to ours is Llark (Gardner et al., 2024). However, neither the model itself nor the music understanding datasets from Llark have been released. None of these models is capable of using music tools, an important ability to interact with creators. In this paper, we propose **OpenMU-Bench** and **OpenMU** to advance the field of music understanding. OpenMU-Bench holistically measures various aspects of music understanding, while OpenMU achieves state-of-art performance on the benchmark. Both OpenMU-Bench and OpenMU are released to facilitate the future research and development in this field.

**Music understanding datasets.** The proliferation of LLMs has spurred the development of benchmarks designed to holistically measure the genuine capabilities of LLMs. Benchmarks have been designed for NLP tasks (BIG-bench authors, 2023; Hendrycks et al., 2021), and vision-language tasks (Liu et al., 2023c; Fu et al., 2023; Ye et al., 2023). MMMU (Yue et al., 2023) included the music modality into evaluation but at a very narrow scope (334 entries of sheet music). Researchers are striving to address the data scarcity challenge of music: Doh et al. (2023) introduced LP-MusicCaps, associating LLM-augmented captions with music clips from MusicCaps (Agostinelli et al., 2023). Similarly, Liu et al. (2024) developed MusicQA, containing QA and captioning tasks for music clips from MusicCaps, MagnaTagATune (Law et al., 2009b), and MTG-Jamendo (Bogdanov et al., 2019). Concurrently, Deng et al. (2024) proposed MusicInstruct, which targets QA and captioning for clips in MusicCaps. Weck et al. (2024) create MuChoMusic as a music understanding benchmark containing 1,187 multiple-choice questions for evaluation. Building on existing datasets, we construct OpenMU-Bench by additionally bootstrapping new datasets using GPT-3.5. OpenMU-Bench contains about one million examples for training and evaluation across various music understanding tasks. We also standardize evaluation metrics to ensure consistency[2] in reporting results on OpenMU-Bench. Table 1 provides the statistics for OpenMU-Bench.

---

[2]For example, we found that MU-Llama (Liu et al., 2024) reports BertScore-Recall, while LP-MusicCaps (Doh et al., 2023) reports BertScore-F1. We standardize the metrics when reporting performance on OpenMU-Bench, and hope this paves the way for consistent evaluations of music understanding MLLMs.

| | Captioning | | Reasoning | | Lyrics | | Tool-Use | | MultipleChoice | | Music Clips |
|---|---|---|---|---|---|---|---|---|---|---|---|
| | Train | Test | Train | Test | Train | Test | Train | Test | Train | Test | |
| **MusicCaps** | 2640 | 2839 | - | - | - | - | - | - | - | - | 5479 |
| **MusicInstruct** | 28670 | 30593 | - | - | - | - | - | - | - | - | (5479) |
| **LPMusicCaps** | 7920 | - | - | - | - | - | - | - | - | - | (2839) |
| **LPMusicMTT** | 51531 | 13386 | - | - | - | - | - | - | - | - | 25863 |
| **Music4all*** | 104268 | 5000 | 449711 | 21543 | - | - | - | - | - | - | 109269 |
| **MusicQA-Fin.** | 31116 | - | 38895 | - | - | - | - | - | - | - | 12543 |
| **MusicQA-Test** | - | 2240 | - | 2800 | - | - | - | - | - | - | 500 |
| **GTZAN*** | 639 | 290 | 2329 | 1116 | - | - | - | - | - | - | 1000 |
| **MusicNet*** | 3791 | 140 | - | - | - | - | - | - | - | - | 330 |
| **MTT*** | - | - | 78839 | 16100 | - | - | - | - | - | - | (25863) |
| **MTG-Jamendo*** | 45129 | 5144 | 177771 | 20308 | - | - | - | - | - | - | 50273 |
| **BART-Fusion** | - | - | - | - | 55262 | 800 | - | - | - | - | 14985 |
| **Tool-Using*** | - | - | - | - | - | - | 1612 | 403 | - | - | 0 |
| **MuChoMusic** | - | - | - | - | - | - | - | - | - | 1187 | 1187 |
| **Total** | 275704 | 54632 | 747545 | 61867 | 55262 | 800 | 1612 | 403 | 0 | 1187 | 221429 |

Table 1: OpenMU-Bench tasks and dataset distributions. "MusicQA-Fin.": MusicQA-Finetuning. *: datasets with our new annotations. Numbers in brackets are not included when calculating the total number of music clips, as they represent captions annotated for the same set of music clips.

## 3 CONSTRUCTING OPENMU-BENCH

This section outlines the construction of OpenMU-Bench. We introduce the five types of tasks included in OpenMU-Bench and explain the dataset construction procedures for each type. In addition to incorporating existing music understanding datasets, we generate new annotations for music clips from datasets that do not contain natural language annotations. Our goal is to integrate as many datasets as possible to enable OpenMU-Bench to comprehensively and systematically evaluate music understanding models. Furthermore, we specify the recommended evaluation metrics to ensure consistent and fair benchmarking. Table 2 shows examples of different OpenMU-Bench task types.

### 3.1 OPENMU-BENCH TASK TYPES

**Music captioning** tasks a model with generating textual descriptions capturing musical contents and key features of a music clip. A music understanding model excels at captioning can improve the efficiency of music production by generating music descriptions in a short time, eliminating the needs of listening to the entire music track by creators. **Music reasoning**, as defined by Gardner et al. (2024), tasks the model with answering questions in two aspects. First, it examines the interaction between different elements of a music clip, such as how a fast tempo is likely to correspond with a high energy level. Second, it explores how the real-world can interact with the music clip, e.g., how a creator can increase the energy level of a music clip by using faster tempos (see Table 2).

**Tool using**. The MIR community has developed a wide range of music technology tools for various tasks, such as tempo estimation, key detection, chord recognition, and instrument identification[3]. Unlike Llark (Gardner et al., 2024), which aims to address many MIR tasks using only the LLM, OpenMU takes a different view. We aim for OpenMU to integrate and leverage the well-established, rigorously tested MIR tools to solve practical, real-world problems. This approach is motivated by the limitations of current LLMs, which often struggle with tasks like arithmetic (Qin et al., 2023). By combining the strengths of established MIR tools (e.g., tempo estimator) with LLM-based method, OpenMU aims to provide a more robust solution to music understanding.

**Lyrics understanding**. Lyrics, which carry rich semantic content, are often used to convey moods and emotions (Watanabe & Goto, 2020). We incorporate a lyrics understanding task into OpenMU-Bench. We consider a model that excels at understanding lyrics to be capable of producing interpretations similar to those of humans, conditioned on the music input. This task is framed as a text generation problem.

**Multiple-choice questions**. Text generation models are inherently difficult to evaluate due to their open-ended nature (Celikyilmaz et al., 2020). This challenge extends to LLM-based music under-

---

[3]List of MIR software tools: https://www.ismir.net/resources/software-tools/

| Music Captioning | **Question**: Write a short description about the musical contents of this clip. |
|---|---|
| | **Answer**: The music clip has a tempo similar to a walking pace and belongs to the blues genre. It features the characteristic blues chord progression and soulful melodies often accompanied by instelements typical of the blues genre are also evident in this clip. |
| Music Reasoning | **Question**: How would a music producer enhance the high energy of this track? |
| | **Answer**: A music producer could enhance the high energy of this track by adding elements like driving basslines, powerful drums, and energetic synthesizers. They could also increase the tempo slightly to create a more intense feel and incorporate dynamic changes in volume and instrumentation to maintain ... |
| Lyrics Understanding | **Question**: Please help me to interpret this lyrics: {muisc_lyrics} |
| | **Answer**: Feel like it's about being depressed about your life. You go to work on Monday, see all of the ... |
| Tool Using | **Question**: What are the chords between 10 sec and 20 sec of this music? |
| | **Answer**: Here are the chords between 10 sec and 20 sec: [GetMusicChords(10, 20)]. |
| Multiple Choice | **Question**: How would you describe the vocal performance in this piece? Options: (A) Soft yet deeply emotional (B) Lamenting (C) Male vocals (D) Operatic |
| | **Answer**: (A) Soft yet deeply emotional. |

Table 2: Example data entires in OpenMU-Bench.

standing models as well. To ensure holistic evaluations, OpenMU-Bench also assesses these models using multiple-choice questions developed by Weck et al. (2024).

## 3.2 INDIVIDUAL DATASETS

As introduced in §2, a few datasets already exist for music understanding. We incorporate these datasets and create new annotations to ensure that OpenMU-Bench has both a large scale and broad coverage. We describe each of the datasets, along with the applied modifications aligning them with the OpenMU-Bench task types. We adhere to existing train/test splits of the datasets when available (c.f. §A.4); Appendix §A.2 details the preprocessing and annotating details of OpenMU-Bench; we highlight only the key information here.

**MusicCaps**, created by Agostinelli et al. (2023), is pivotal for the music captioning task. It contains approximately 5.5K 10-second music clips sourced from AudioSet (Gemmeke et al., 2017), with corresponding gold-standard text captions written by professional musicians. We incorporate MusicCaps into OpenMU-Bench as part of the captioning task. **LPMusicCaps & LPMusicMTT** (Doh et al., 2023) extend MusicCaps and the MagnaTagATune (Law et al., 2009a) dataset by generating additional textual descriptions. The authors prompt GPT-3.5 to "write", "summarize", "paraphrase", and "predict attributes" new captions to the music clips. We integrate[4] approximately 8K LPMusicCaps and 51K LPMusicMTT training captions into OpenMU-Bench. **MusicInstruct** (Deng et al., 2024) also extends MusicCaps by creating question-answer pairs for the MusicCaps clips using GPT-4. This dataset contains approximately 60K question-answer pairs, which are categorized into two versions: a short version (**MI-short**) focusing on musical content such as tempo and genre, and a long version (**MI-long**) that paraphrases the MusicCaps captions. We integrate MusicInstruct into OpenMU-Bench as a captioning task, and report performance on both versions separately.

**MusicQA**, developed by Liu et al. (2024) by prompting MPT (MosaicML-NLP-Team, 2023), is employed to train their MU-LLaMA. MusicQA is composed of MusicCaps clips for pretraining, MagnaTagATune (Law et al., 2009a) clips for finetuning, and MTG-Jamendo (Bogdanov et al., 2019) clips for testing. We incorporate MusicQA-Finetune and MusicQA-Test into OpenMU-Bench, while MusicQA-Pretrain, which contains the test split of MusicCaps, is excluded to prevent potential train-test leakage (Deng et al., 2024). Following Liu et al. (2024), we separate MusicQA into captioning and reasoning parts.

**Music4all**, developed by Pegoraro Santana et al. (2020), consists of approximately 100K music clips with rich metadata, including attributes like energy, valence, and genre. Based on this metadata, we prompt GPT-3.5 to generate annotations for both the captioning and reasoning tasks. The prompts used for these annotations are provided in the Appendix §A.3. **GTZAN**, developed by Tzanetakis & Cook (2002), contains approximately 1K 30-second music clips, each labeled with genre tags and we create extra tempo tags with Madmom (Böck et al., 2016). Based on these tags, we generate captioning and reasoning annotations with prompting. **MusicNet** (Thickstun et al., 2017) contains 1

---

[4]We do not use the "attribute prediction" annotations, following the recommendation from the LPMusic authors: https://huggingface.co/datasets/seungheondoh/LP-MusicCaps-MC

million dense annotations at precise timestamps for 330 classical music recordings. The annotations are of high quality, but primarily focus on instruments. As a result, we integrate MusicNet into OpenMU-Bench as part of the captioning task, retaining only annotations that span three seconds or longer. **MagnaTagATune** (MTT) has been included in OpenMU-Bench as part of the captioning task, thanks to the annotations by Doh et al. (2023). Given its significance in the MIR community (Won et al., 2020), we also create an additional 90K reasoning annotations for training and testing. **MTG-Jamendo** (Bogdanov et al., 2019) consists of approximately 55K full music tracks, each tagged with genre, instrument, and mood. We randomly select 30-second music clips[5] and generate annotations for captioning and reasoning tasks by prompting GPT-3.5.

**Tool using**. To the best of our knowledge, there is no existing dataset designed to train models in leveraging MIR tools. To address this, we generate training and testing datasets for solving four MIR tasks with tools: chord recognition, tempo estimation, key detection, and downbeat extraction. We demonstrate that OpenMU quickly learns to utilize these tools to answer queries related to MIR information. We implemented these tools by wrapping the Python package `Madmom` (Böck et al., 2016); §A.5 shows implementation details.

For **Lyrics understanding**, we integrate BART-fusion (Zhang et al., 2022)'s annotations, containing internet interpretations to the lyrics and music clips of Music4all. For **Multiple-choice questions**, we integrate MuChoMusic (Weck et al., 2024) for evaluation. The task involves answering questions about music knowledge and reasoning by selecting the correct option from four provided choices.

### 3.3 EVALUATION METRICS

OpenMU-Bench leverages common evaluation metrics for text generation tasks: captioning, reasoning, and lyrics understanding. BLEU-1, BLEU (Papineni et al., 2002)[6], Meteor (Banerjee & Lavie, 2005), Rouge-1, and Rouge-L (Lin, 2004) measure an answer's textual overlap with the gold standard, while BertScore (Zhang et al., 2020) measures similarity in the semantic representation space of a pretrained BERT model. For all evaluations, we report the scores computed using the F-measure. We report accuracy for the task of multiple-choice questions.

## 4 MODEL ARCHITECTURE AND TRAINING DETAILS

### 4.1 MODEL ARCHITECTURE

**Encoding music clips.** We use AudioMAE (Huang et al., 2022) to encode an input music clip into vector representations. Specifically, we use the "ViT-B AS-2M pretrained + finetuned" version of AudioMAE, which is a Vision Transformer (Dosovitskiy et al., 2021) initially pretrained with a masked auto-encoding reconstruction loss (He et al., 2022), followed by finetuning on tagging tasks (Gemmeke et al., 2017), both using the AudioSet2M (Gemmeke et al., 2017) dataset. The choice of using AudioMAE over other music encoders, such as MERT (Li et al., 2024) or Jukebox-5B (Dhariwal et al., 2020; Castellon et al., 2021), is motivated by two primary reasons. First, more than half of the audio clips inAudioSet2M[7] consist of music or musical instrument recordings, resulting in approximately 3,137 hours of music data (compared to the 910 hours in the MERT-95M-public model (Li et al., 2024)). Audio encoders pretrained on AudioSet have shown competitive performance in music tagging tasks (Koutini et al., 2021; Niizumi et al., 2022). Second, the size of the multimodal encoder is not a performance bottleneck (McKinzie et al., 2024). Instead, the smaller number of parameters in ViT-B (86M) facilitates more efficient training.

**LLM**. We use the open-sourced Llama3-8B-instruct (Dubey et al., 2024) as our LLM. Compared to previous Llama models (Touvron et al., 2023), Llama3 has been trained on higher-quality datasets and at larger scales, achieving GPT-4-level performance (Achiam et al., 2023) on numerous tasks.

**Music-language projector** links the representation space of the music encoder with the LLM. Studies (McKinzie et al., 2024; Liu et al., 2023a) have shown that the architecture of the projector itself has little impact on downstream task performance, while the number of tokens from the multimodal

---

[5]We provide scripts for extracting music clips identical to ours.

[6]Following the machine translation literature, our BLEU refers to BLEU-4.

[7]AudioSet2M Ontology: `https://research.google.com/audioset/`

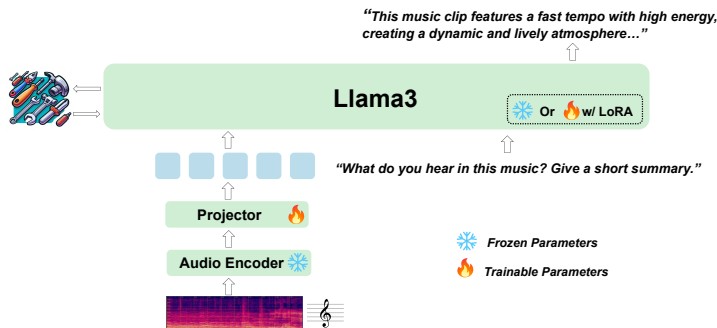

Figure 1: Model architecture of OpenMU. In Stage (1), we only tune the music-language projector. In Stage (2), LoRA adapters are added to the LLM and are tuned together with the projector.

encoder is significantly important. We use a two-layer MLP with GELU non-linearity (Hendrycks & Gimpel, 2017) and evaluate the effect of varying the number of music tokens in §5.1.

Overall, OpenMU follows the well-tested architecture of MLLMs (McKinzie et al., 2024) as shown in Figure 1. In contrast to previous music MLLMs such as MU-LLaMA, OpenMU is also capable of interacting with external MIR tools such as tempo estimator (Böck et al., 2016).

## 4.2 TRAINING DETAILS

**Dataset preprocessing**. When processing the music clips, we limit their maximum length to 30 seconds and zero-pad those shorter than 30 seconds. All music clips are resampled to 16 kHz and then converted to a 128-bin Mel-spectrogram with a 25-ms hann window and 10-ms hop size. Consequently, each music clip is represented as a mel-spectrogram with a shape of (3072, 128). Since AudioMAE is trained to encode inputs of up to 10 seconds, we segment each mel-spectrogram into three parts, encode them separately, and then concatenate the results. As a result, each 30-second music clip is encoded by 1536 tokens, with each token having a shape of (1, 768).

Throughout our experiments, we used between 8 and 16 A100 40GB GPUs, depending on the experimental setup (c.f. §5). In all experiments, we set the maximum context length of the LLM to 2048 tokens. We utilized DeepSpeed ZeRO-3 (Rajbhandari et al., 2020) and FlashAttention2 (Dao et al., 2022) to enable fast and efficient training. It took approximately three days to train OpenMU on the captioning and reasoning subsets of OpenMU-Bench (around 1 million data examples).

**Training setup** of OpenMU-Bench largely follows the common practice of MLLM training (Yin et al., 2023; Liu et al., 2023b; McKinzie et al., 2024), consisting of:

**Stage (1) Captioning**. We train OpenMU to generate captions, conditioned solely on the input music clip. The goal of Stage (1) training is to align the representation spaces of AudioMAE and Llama3, with the only trainable module in this stage being the music-language projector. We use the captioning subset of OpenMU-Bench for training in this stage. A key configuration is the number of music tokens fed into the LLM, which we discuss in detail in §5.1. The remaining hyperparameters largely follow Liu et al. (2023b) and are provided in the Appendix §A.1.

**Stage (2) Instruction Tuning**. After aligning the music and text representation spaces, Stage (2) training enables OpenMU to follow various instructions in the music domain, such as inferring music genres or reasoning about the content of a music clip. In this stage, LoRA adapters (Hu et al., 2022) are incorporated into OpenMU's LLM, followed by fine-tuning on OpenMU-Bench's captioning and reasoning tasks. We focus on two critical research questions in this stage. First, we extensively evaluate OpenMU's task performance with respect to its LoRA parameters (see §5.2). Second, we investigate in-depth OpenMU's use of music information. Given the large-scale pretraining data of OpenMU's LLM, we hypothesize that OpenMU might be able to make correct predictions for knowledge-intensive questions even without relying on musical information within a music clip. We test this hypothesis and show that in order to achieve higher performance, OpenMU indeed relies on information from the music clip, demonstrating OpenMU's genuine ability to understand music.

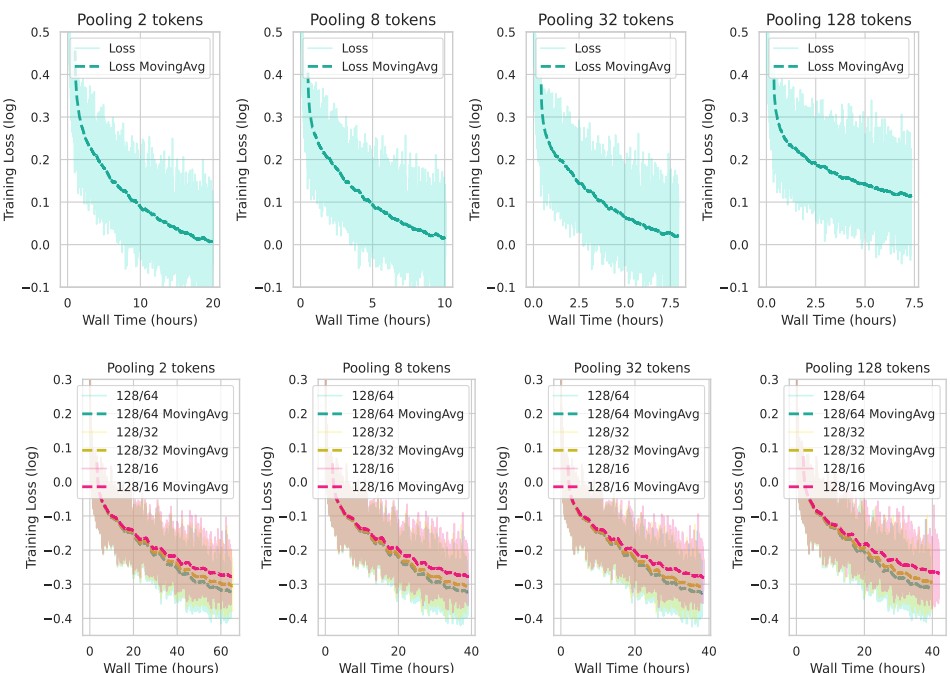

Figure 2: Training trajectories of Stage (1) (top) and Stage (2) (bottom). The x-axis represents the number of hours elapsed, and the y-axis shows the training loss on a log scale. We vary the number of mean-pooling music tokens from 2 to 128 and experiment with different LoRA parameter combinations, $\alpha/r$. "MovingAvg" represents the moving average.

## 5 DESIGNING OPENMU AND DISCUSSIONS

In this section, we explore and discuss the critical factors involved in training OpenMU. We aim for these detailed analyses to contribute to the research and development of future foundation models for music understanding.

### 5.1 NUMBER OF MUSIC TOKENS

McKinzie et al. (2024) illustrate that the number of image tokens is more significant than the architecture of the vision-language projector in vision-language MLLMs. To the best of our knowledge, no prior research has addressed this critical aspect in the context of training foundation models for music understanding. This is particularly important because music clips can often be lengthy, leading to a large number of music tokens. For instance, the AudioMAE encoder outputs 1536 tokens for representing a 30-second music clip. While using all available tokens ensures the maximum use of music modality information, it may hinder training efficiency and limit the utility of the context window of the LLM (2048 in our case). In this section, we extensively evaluate the impact of the number of music tokens when training OpenMU.

Figure 2 displays the training trajectories (log-scale) of both Stage (1) and Stage (2) training, where we apply mean-pooling to every 2–128 music tokens output by AudioMAE. For instance, mean-pooling every 8 tokens means using only 1536/8 = 192 tokens to represent the 30-second input music clip. The difference among mean-pooling 2, 8, and 32 tokens is small, suggesting that there may be *redundancies in the representations of the encoded music clip.* Although aggressively mean-pooling 128 tokens significantly reduces the overall training time (7.5 hours when pooling 128 tokens vs. 20 hours when mean-pooling 2 tokens), the setting results in a weaker convergence. As a result, we empirically focus on model variants with mean-pooling 8 tokens in the next sections to balance between model convergence and training efficiency.

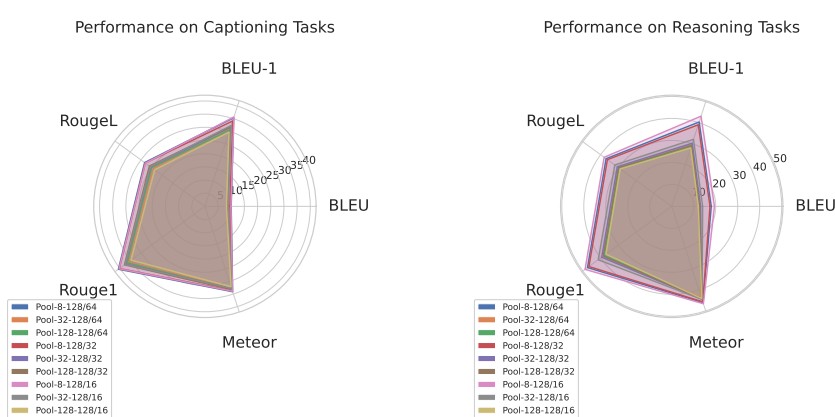

Figure 3: Performance of OpenMU variants on the captioning and reasoning tasks of OpenMU-Bench. For each evaluation metric, such as BLEU, we report the macro average of the model's performance across all OpenMU-Bench subtasks.

## 5.2 LoRA, TASK PERFORMANCE, AND MUSIC INFORMATION UTILITY

As introduced in §4.2, Low-Rank Adaptation (LoRA; Hu et al. (2022)), is employed in Stage (2) training to efficiently adapt OpenMU's LLM for following instructions in the music domain. Given an LLM weight parameter matrix $W \in \mathbb{R}^{d \times k}$, instead of directly modifying $W$, LoRA introduces and trains two matrices, $B \in \mathbb{R}^{d \times r}$ and $A \in \mathbb{R}^{r \times k}$ for adapting $W$ to a downstream task:

$$W \leftarrow W + \frac{\alpha}{r} BA.$$

The LLM weight matrix $W$ remains unchanged; the LoRA rank $r$ determines the number of trainable parameters by controlling the size of $B$ and $A$. The matrix multiplication result, $BA$, represents the changes introduced by adaptation to a downstream task, scaled by $\frac{\alpha}{r}$. Here, $\alpha$ is a hyperparameter, and typically $r < \alpha$. For OpenMU, we fix $\alpha = 128$ following Liu et al. (2023b;a), while varying the value of $r$. Intuitively, a smaller rank $r$ imposes a stricter bottleneck on $B$ and $A$, requiring the learned parameter differences, represented by $BA$, to rely on fewer trainable parameters to capture concise and genuine information about the downstream task, which are subsequently scaled by a larger $\frac{\alpha}{r}$. In contrast, a larger $r$ introduces more trainable parameters, which may be prone to learning shortcuts, redundant information, or noise (Geirhos et al., 2020) during adaptation to the downstream task, subsequently scaled by a smaller $\frac{\alpha}{r}$. In this section, we investigate how LoRA configurations affect Stage (2) training, as well as reporting the evaluation results of OpenMU on OpenMU-Bench.

**OpenMU-Bench task performance.** Figure 3 shows the performance of OpenMU variants on the captioning (left) and reasoning (right) tasks of OpenMU-Bench. For each evaluation, we report the macro-average performance of each OpenMU variant across all subtasks in OpenMU-Bench. Additionally, Figure 4 (left) displays the evaluation results using BertScore (Zhang et al., 2020) as the metric. Several observations can be made. First, the number of music tokens plays a critical role in task performance, echoing the conclusion drawn by McKinzie et al. (2024). Mean-pooling every 8 tokens shows clear advantages over 32 and 128 tokens, likely due to its preservation of music information. However, mean-pooling 32 tokens offers only limited improvement over 128 tokens, and the performance decline appears to plateau. It is likely that crucial music information is already lost when mean-pooling 32 tokens. Second, the effectiveness of LoRA parameters show limited impacts on task performance, similar to the findings in Gong et al. (2024). *As a result, we will focus on the model variant with mean-pooling 8 tokens, and LoRA parameters 128/16 in the next sections.*

**Music information utility.** Given the large-scale pretraining data of the LLM, which already contains rich knowledge about music, an MLLM may be able to answer questions about music without relying on the information in the music clip. Hence, this section addresses a key question: Does OpenMU genuinely utilize information from the input clip to understand the music more effectively? To investigate this, we evaluate OpenMU variants on MuChoMusic (Weck et al., 2024), a dataset containing multiple-choice questions focused on music understanding. Questions such as "Which sub-genre of rock music would best classify this piece?" require the model to select the

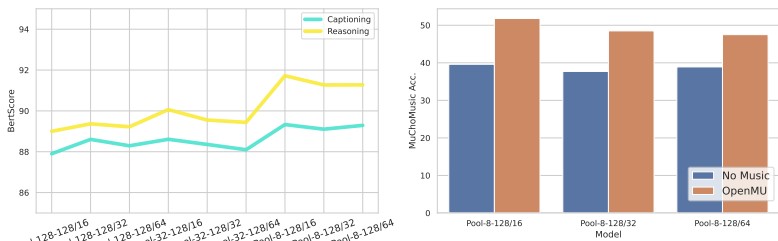

Figure 4: Left: Performance of OpenMU variants on the captioning and reasoning tasks of OpenMU-Bench using BertScore as the metric. Right: OpenMU performance on MuChoMusic.

| | BLEU-1 | | BLEU | | Rouge1 | | RougeL | | BertScore | | Meteor | |
|---|---|---|---|---|---|---|---|---|---|---|---|---|
| | OMU | MUL | OMU | MUL | OMU | MUL | OMU | MUL | OMU | MUL | OMU | MUL |
| **MusicCaps** | **25.62** | 9.78 | **2.89** | 0.73 | **27.99** | 21.22 | **18.56** | 15.64 | 86.63 | **86.85** | **21.83** | 11.03 |
| **MI-short** | 18.92 | **50.01** | 8.75 | **23.90** | 41.03 | **52.69** | 37.75 | **47.73** | 89.90 | **92.83** | 43.81 | **49.10** |
| **MI-long** | **36.66** | 2.13 | **4.18** | 0.18 | **38.74** | 19.34 | **21.70** | 13.55 | **87.24** | 85.90 | **24.43** | 8.83 |
| **LPMusicMTT** | **23.83** | 18.87 | **2.56** | 0.56 | **29.23** | 21.68 | **21.57** | 16.07 | **89.75** | 88.37 | **24.60** | 13.96 |
| **Music4all**\* | **51.03** | 5.11 | **18.69** | 0.36 | **51.31** | 19.51 | **34.80** | 14.31 | **91.04** | 86.64 | **43.58** | 10.07 |
| **MusicQA-Test** | 19.60 | **19.65** | 2.22 | **5.08** | 23.96 | **31.74** | 17.39 | **28.07** | 87.30 | **89.45** | 26.34 | 21.73 |
| **GTZAN**\* | **45.38** | 4.56 | **11.78** | 0.32 | **44.01** | 20.41 | **27.62** | 15.17 | **89.34** | 87.03 | **34.71** | 10.65 |
| **MusicNet**\* | **52.68** | 1.17 | **22.14** | 0.00 | **56.44** | 14.49 | **38.11** | 12.25 | **91.97** | 85.59 | **45.92** | 6.56 |
| **MTG-Jamendo**\* | **47.56** | 5.32 | **15.83** | 0.41 | **49.66** | 20.29 | **33.76** | 15.94 | **90.79** | 87.72 | **39.44** | 10.06 |

Table 3: OpenMU-Bench captioning results (in %) of OpenMU (OMU) and MU-LLaMA (MUL).

correct option from four candidates. Notably, such questions could be answered based on the most common or probable sub-genre from the LLM's pretraining data, allowing the model to perform reasonably well without actually relying on the music input. Figure 4 (right) presents the MuChoMusic results of OpenMU variants. The "No Music" condition refers to replacing the input music clip with a white noise clip, while "OpenMU" displays the results when actual music clips are used. It is evident that music information is crucial for OpenMU to achieve strong performance; OpenMU effectively utilizes music information rather than relying on shortcuts (Geirhos et al., 2020).

## 6 OVERALL RESULTS

In this section, we compare OpenMU with MU-LLaMA, a widely used music understanding model, on OpenMU-Bench. For OpenMU, we use the variant of mean-pooling 8 tokens with LoRA parameters 128/16. For MU-LLaMA, we use the checkpoint released by Liu et al. (2024).

**Music captioning and reasoning** results are presented in Table 3 and Table 4, respectively. We observe that OpenMU consistently outperforms MU-LLaMA across various captioning and reasoning tasks. Interestingly, MU-LLaMA lags behind OpenMU on MusicCaps, despite the fact that the MusicCaps test set was used during MU-LLaMA's pretraining stage (Liu et al., 2024; Deng et al., 2024). We believe this is due to the small size of MusicCaps—its effectiveness was likely overshadowed by the larger finetuning datasets used for MU-LLaMA.

MU-LLaMA outperforms OpenMU on the MusicInstruct-short captioning task (Deng et al., 2024) and the MusicQA-test reasoning task (Liu et al., 2024) in terms of surface form matching metrics by a large margin. However, we found that the gold references in these two subsets are biased to contain

| | BLEU-1 | | BLEU | | Rouge1 | | RougeL | | BertScore | | Meteor | |
|---|---|---|---|---|---|---|---|---|---|---|---|---|
| | OMU | MUL | OMU | MUL | OMU | MUL | OMU | MUL | OMU | MUL | OMU | MUL |
| **Music4all**\* | **49.20** | 18.13 | **23.31** | 5.97 | **53.26** | 34.11 | **41.08** | 25.01 | **92.34** | 89.63 | **49.96** | 22.51 |
| **MusicQA-Test** | 24.84 | **40.64** | 9.46 | **22.47** | 35.86 | **51.29** | 30.66 | **47.54** | 89.70 | **92.59** | 40.04 | **46.15** |
| **GTZAN**\* | **50.26** | 16.16 | **22.07** | 5.95 | **52.96** | 35.18 | **38.57** | 26.20 | **92.02** | 89.89 | **46.87** | 21.84 |
| **MTT**\* | **45.52** | 21.70 | **21.18** | 8.31 | **50.83** | 38.70 | **39.93** | 29.92 | **92.03** | 90.63 | **48.06** | 26.33 |
| **MTG-Jamendo**\* | **45.87** | 23.69 | **21.12** | 8.59 | **50.78** | 38.45 | **39.74** | 29.27 | **92.01** | 90.47 | **47.97** | 26.42 |

Table 4: OpenMU-Bench reasoning results (in %) of OpenMU (OMU) and MU-LLaMA (MUL).

|  | Accuracy | | | IFR |
|---|---|---|---|---|
|  | All | Knowledge | Reasoning | All |
| **MusiLingo** | 21.1 | 22.0 | 19.2 | 71.6 |
| **MU-LLaMA** | 32.4 | 32.3 | 31.3 | 79.4 |
| **M2UGen** | 42.9 | 44.9 | 41.2 | **96.4** |
| **OpenMU** | **51.8** | **51.4** | **51.4** | 94.8 |
| Random | 25.0 | 25.0 | 25.0 | 100.0 |

Table 5: MuChoMusic accuracy and instruction-following rate (IFR) of OpenMU and prior music understanding models. Numbers are in %. MuChoMusic contains multiple-choice questions; MusiLingo, MU-LLaMA, M2UGen performances are from Weck et al. (2024). "Random" shows random guessing results. We assume "Random" will always select an option, hence its IFR is 100%.

a large portion of repeated parts from the questions. For example, in MusicQA-test reasoning, to the question "What is the alternative genre of music in the audio?", the gold standard reference is "The alternative genre of music in the audio is postrock." This observation is further supported by objective metrics. Compared to the edit distance of 225 and the Jaccard similarity score of 23.9% for MTT, MusicQA-test reasoning has an edit distance of 90 and a Jaccard similarity score of 36.2%. Since MU-LLaMA tends to repeat the question before providing an answer, the behavior might have inflated the surface-level form matching scores in these subsets. As a result, we recommend practitioners downweight these subsets when evaluating music understanding models.

|  | BLEU-1 | BLEU | Rouge1 | RougeL | BertScore | Meteor |
|---|---|---|---|---|---|---|
| BART-Fusion | **25.79** | **6.48** | **32.18** | **17.99** | 83.03 | **27.97** |
| OpenMU | 25.60 | 5.19 | 31.31 | 17.03 | **83.14** | 27.01 |

| Chords | 94.95 |
|---|---|
| Tempo | 95.83 |
| Key | 100 |
| Downbeats | 100 |

Table 6: Lyrics understanding results (left) and tool using accuracy (right). Numbers are in %.

**Multiple-choice questions**. We compare OpenMU with MU-LLaMA, along with other available music understanding models, on the multiple-choice question dataset MuChoMusic. Deng et al. (2024) is a concurrent work to MU-LLaMA while M2UGen (Hussain et al., 2023) adds music generation ability to MU-LLaMA. Table 5 shows that OpenMU achieves state-of-the-art music understanding performance on MuChoMusic.

**Lyrics understanding.** Table 6 (left) compares OpenMU's performance with BART-fusion (Zhang et al., 2022), a model specifically designed for lyrics understanding. For simplicity, we reuse the same hyperparameters from Stage (2) training, except for extending the training to 20 epochs. OpenMU outperforms BART-fusion in BertScore but slightly lags behind on other metrics. Future models could explore further hyperparameter tuning or architectural modifications to improve performance. **Tool using accuracy**. Table 6 (right) reports the accuracy of OpenMU when calling external MIR tools. We consider an exact match as a hit. For example, in chords estimation, if the gold reference is "[GetMusicChords(10, 20)]", the model must accurately output the type and arguments of the tool call to be considered a hit. Extra calls are considered a miss. As expected, OpenMU performs well on this task and learns to call MIR tools effectively. It is promising to integrate more MIR tools to handle a broader range of task types and complexities.

## 7 CONCLUSION

We presented OpenMU-Bench, a large-scale benchmark suite containing approximately one million examples for training and evaluating LLM-based music understanding models. We construct OpenMU-Bench by creating new annotations as well as leveraging existing datasets. We trained our music understanding model, OpenMU, with extensive ablations and demonstrated that it outperforms baseline models such as MU-LLaMA. Both OpenMU and OpenMU-Bench are open-sourced to facilitate future research in music understanding and enhance the efficiency of creative music production. Future work may explore extending OpenMU to support multiple music clips as input and enable in-context learning for music understanding. Another promising direction is enabling OpenMU to integrate more MIR tools, combining the strengths of LLMs and established tools for deeper music understanding.

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

| | Tempo | Energy | Valence | Danceability | Genre | Mood | Instrument | Others |
|---|---|---|---|---|---|---|---|---|
| **Music4all** | ◯ | ◯ | ◯ | ◯ | △ | △ | △ | △ |
| **GTZAN** | ◯ | × | × | × | ◯ | × | × | × |
| **MusicNet** | × | × | × | × | × | × | ◯ | × |
| **MTT** | × | × | × | × | △ | △ | △ | △ |
| **MTG-Jamendo** | × | × | × | × | △ | △ | △ | △ |

Table 7: The metadata associated with each subtask dataset in OpenMU-Bench. ◯: the metadata, e.g., tempo, is available for the music clips. △: the metadata *maybe* available for some music clips, but not for all of them. ×: the metadata is not available for the music clips.

## A  APPENDIX

### A.1  TRAINING DETAILS AND HYPERPARAMETERS

In this section, we describe the detailed settings and hyperparameters used for training OpenMU.

All experiments were conducted using 8-16 A100 40GB GPUs, with BF16 enabled to ensure stable training. We use DeepSpeed ZERO-3 (Rajbhandari et al., 2020) and Flash Attention 2 (Dao et al., 2022) to reduce the memory consumption. We utilized the Adam optimizer and a cosine learning rate scheduler, with a 30% warm-up ratio.

For Stage (1) training, we pretrained OpenMU for 15 epochs on the captioning subtask of OpenMU-Bench, which consists of approximately 275K pairs of music clips and corresponding captions. Stage (1) training took approximately 10 hours for the checkpoint we evaluated (i.e., mean-pooling every 8 music tokens as illustrated in Figure 2. The initial learning rate was set to 1e-3, with a batch size of 8 per GPU.

For Stage (2) training, we extended pretraining of OpenMU for 10 epochs on the captioning and reasoning subtasks of OpenMU-Bench, which comprise roughly one million training examples. The initial learning rate was set to 2e-5, with the same per-GPU batch size of 8. Stage (2) required approximately 40 hours due to the increased size of training data.

For the lyrics understanding subtask, we trained OpenMU for 20 epochs, reusing the hyperparameters from Stage (2). Similarly, for the tool using subtask, we reused the Stage (2) hyperparameters but reduced the number of epochs to 5 due to the smaller dataset size for this task.

### A.2  METADATA OF DATASETS

In this paper, we contribute to creating the large-scale benchmark suite OpenMU-Bench for music understanding.

In contrast to other modalities such as images, where rich natural language descriptions are widely available across the internet (Schuhmann et al., 2022), music clips are often accompanied by tags, such as genre, year, and instruments. We consider these tags to be a form of metadata for the music clips. When constructing OpenMU-Bench, we bootstrap captions and reasoning texts in natural language about the music clips based on this metadata by prompting GPT-3.5.

Table 7 demonstrates the various types of metadata used in the OpenMU-Bench subtasks to create music understanding examples. Due to the broad coverage, music clips from different OpenMU-Bench subsets are associated with diverse types of metadata. Even within the same subtask, different music clips may be tagged with only a limited set of metadata types. We detail how we process the metadata of each music clip as follows.

**Tempo**. Music clips in two datasets, music4all and GTZAN, are associated with tempo, and we convert the numerical values into natural language descriptions, following the Italian musical terms[8] as shown in image Figure 5.

For **energy**, **valence**, **danceability**, which are float scores ranging from 0 to 1, we convert them into natural language descriptions using empirical thresholds of 0.3 and 0.7. Taking energy as an

---

[8]Italian musical terms: https://www.musicca.com/musical-terms.

| Very Slow | Slow | Walking Pace | Medium | Fast | Very Fast | Extremely Fast |
|-----------|------|--------------|--------|------|-----------|----------------|

45BPM          76 BPM          108 BPM     116 BPM     168 BPM     200 BPM

Figure 5: Converting numerical values (in beats per minute; BPM) of music tempo to natural language descriptions. The conversion is done based on the Italian musical terms.

example, we consider a energy level $s$, where $s \geq 0.7$ as a high energy level, $0.7 > s \geq 0.3$ as a medium energy level, and $0.3 > s$ as a low energy level.

For **genre**, **mood**, and **instrument** of Music4all, MTT, MTG-Jamendo, we merge their original metadata by manual annotations and corrections for consistency. Concretely, we keep the top 50 tags of MTT and MTG-Jamendo, following the recommendation of the authors (Law et al., 2009b; Bogdanov et al., 2019), and use the top 166 tags of Music4all, as recommended by `music4all_contrib`[9]. The corrections involves actions such as de-compounding ("acoustic-guitar" → "acoustic guitar"), unifying ("Female vocalists" → "female vocal"), expanding ("synth" → "synthesizer") the tags, and the resulting metadata tags are list as follows:

---

**Metadata of genre, instrument, mood, and others**

```
Genre:
    instrumental, triphop, world, pop punk, hardcore, metalcore, mb, 70s,
    death metal, dream pop, brazilian, easylistening, classical, metal,
    rock, instrumental pop, 90s, dance, reggae, acoustic, 80s, orchestral,
    lounge, indie pop, british, electronic, pop, soul, experimental,
    hip-hop, indian, indie, indie rock, heavy metal, 60s, punk,
    progressive rock, synthpop, jazz, hard rock, post-hardcore, funk,
    alternative rock, new age, post-punk, pop rock, trance, mpb, pop folk,
    classic rock, techno, soundtrack, new wave, atmospheric, country,
    lo-fi, downtempo, rap, folk, opera, house

Instrument:
    harpsichord, piano, strings, choral, flute, vocal, keyboard, violin,
    drums, computer, bass, harp, drum machine, acoustic guitar, electric
    guitar, no vocal, electric piano, synthesizer, cello, female vocal,
    guitar, male vocal, sitar

Mood:
    psychedelic, soft, energetic, film, weird, ambient, loud, slow,
    chillout, relaxing, quiet, fast, happy, emotional

Others:
    beats, solo, singer-songwriter
```

---

As a result, a JSON formatted metadata is created for each of the music clips:

```
{
    "dataset_name": "music4all",
    "audio_filename": "4MqXFtyr1XwxrShX.mp3",
    "tempo": "walking pace tempo",
    "valence": "medium valence",
    "energy": "high energy",
    "danceability": "medium danceable",
    "genre": [
      "rock",
      "pop",
      "electronic"
    ],
    "mood": [
      "ambient"
    ]
}
```

---

[9]https://github.com/keunwoochoi/music4all_contrib

which is then employed to prompt GPT3.5 to create examples for music understanding, as described in the next section §A.3.

## A.3 PROMPTS AND DATASET FORMAT

Based on the metadata of each music clip (§A.2), we prompt GPT-3.5 to generate examples for the music understanding tasks. Our prompts are adapted from those used by Gardner et al. (2024), with modifications tailored to the available metadata of different OpenMU-Bench subsets and subtasks. As an example, we present the prompt used for the music captioning task in Music4All:

---

**Example prompt used for Music4all**

```
You are an expert AI assistant that is knowledgeable about music
production, musical structure, music history, and music styles, and
you are hearing audio of a short clip or loop extracted from a piece
of music. What you hear is described in a JSON-format shown below,
describing the same audio clip you are listening to. This description
is provided in a JSON dictionary, where the keys and values represent
attributes of the music clip.  The JSON contains a set of fields
describing various features of the music clip.

Specifically, the JSON will contain:
    - "tempo": the tempo of this music clip.
    - "energy": the energy level of the music clip. High energy means
      fast, loud, and noisy. For example, death metal has high energy,
      while a Bach prelude scores low on energy. Perceptual features
      contributing to this attribute include dynamic range, perceived
      loudness, timbre, onset rate, and general entropy.
    - "valence": the valence level of the music clip. Valence measures
      the musical positiveness conveyed by the song. High valence
      value means the song is more positive (e.g. happy, cheerful,
      euphoric), otherwise the song is more negative (e.g. sad,
      depressed, angry).
    - "danceability": danceable level of this music clip. Danceability
      measures how suitable a song is for dancing based on a
      combination of musical elements including tempo, rhythm
      stability, beat strength, and overall regularity.

Optionally, the JSON may contain the following OPTIONAL annotations of
the song:
    - "genre": a list of genres of the music clip.
    - "mood": a list of possible moods of the music clip.
    - "instrument": a list of possible instruments used by the music
      clip.
    - "others": other informative factors of the music clip.

You should only use the optional annotations when provided. Otherwise,
you MUST NOT mention them in the description.

Based on the JSON, **provide a detailed musical description of the
clip, from the perspective of a musical expert describing the clip as
they hear it being played**. Use your knowledge about music to make
sure to describe the musical style and contents, and any unique
features of the music clip.

Describe the music clip using a tone of describing facts. DO NOT USE
languages such as "I hear ...". Just provide musical captions to the
clip.

Only provide details that are based on the provided metadata or your
background knowledge of music as an intelligent AI assistant. Explain
any musical concepts that would be unfamiliar to a non-musician. Do
not specifically reference the provided metadata in the response;
instead, respond as if you are hearing the song and reporting a rich
description of what you hear. Keep in mind that the music clip is only
a short clip, loop, or part of a song, and NOT THE COMPLETE SONG.

IMPORTANT!! DO NOT use the word "metadata" anywhere in your
descriptions. DO NOT disclose that metadata about the song is provided
to you. DO NOT use the phrase "song data" anywhere. Do not reveal that
you know details of how the music clip was produced; instead, use
phrases like "it sounds like XXX instrument" or "what I hear might be
a YYY microphone". Now please provide the musical description of the
clip, strictly following above requirements.
```

---

In short, we ask GPT-3.5 to generate an informative caption that can describe the key features and components of the music clip, based on its internal music knowledge accumulated during pretraining and our provided rich metadata. The generated results of the example in §A.2 is:

```
{
    "instruction": "Explain the contents of this song.",
    "output": "This music clip features high energy with a
        walking pace tempo. It combines elements from rock,
        pop, and electronic genres, creating a dynamic and lively
        soundscape. The medium valence contributes to a balanced
        emotional tone, while the medium danceability suggests a
        rhythmic and engaging quality, suitable for movement and dance.
        The ambient mood adds a layer of atmospheric texture to the
        overall vibrant and energetic composition.",
    "local_audio_path": "4MqXFtyr1XwxrShX.mp3",
    "task": "captioning",
    "dataset": "music4all_test"
},
```

which is then leveraged to train or test OpenMU according to the dataset split.

### A.4 DATASET SPLITS

In this section, we provide details about the train/test splits of the OpenMU-Bench subtasks. Specifically, for MusicCaps, MusicInstruct, LPMusicCaps, LPMusicMTT, MusicQA, MusicNet, BART-Fusion, and MuchoMusic, we follow the train/test splits proposed in the original papers. For GTZAN, we used the widely accepted filter-fault split (Kereliuk et al., 2015), and the split from MARBLE (Yuan et al., 2023) for MTT.

For Music4All, we start with the 800 music clips from BART-Fusion as the initial test set. We then expand this set by randomly sampling music clips until the total reaches 5,000. The remaining music clips and their annotations are used as training data. For MTG-Jamendo, we use annotations where the music clips from folds 90 to 99 of the original dataset (Bogdanov et al., 2019) serve as the test data, while the remaining clips and their annotations are treated as training data. For tool using, we randomly sample 80% examples for training and 20% for testing.

### A.5 TOOLS

We define simple tools for solving MIR tasks such as tempo estimator. They are implemented as simple Python wrapper to the `Madmom` toolkit (Böck et al., 2016), which has been widely used in MIR. For example, the tempo estimator can be implemented[10] as:

```
from madmom.features.beats import RNNBeatProcessor
from madmom.features.tempo import TempoEstimationProcessor

def EstimateTempo():
    wav = load_audio(AUDIO_FILE)
    beat_proc = RNNBeatProcessor()
    tempo_proc = TempoEstimationProcessor(fps=100)
    beat_acts, tempo_acts = beat_proc(wav), tempo_proc(beat_acts)
    tempo_est = round(tempo_acts[0][0], 1)
    return tempo_est
```

OpenMU then calls for such a tool to estimate the tempo, when being asked questions such as "Let me know the tempo of this music clip." and replying with "The music has tempo [EstimateTempo() $\rightarrow n$] beats per minute.".

---

[10]In our implementation, we use pseudo names, such as `F1`, for the tools. We found that Llama3 tends to hallucinate new tools when camel case names like "EstimateTempo" are used, likely due to the presence of code in its pretraining data (Dubey et al., 2024).

