# OpenReview forum: "OpenMU: Your Swiss Army Knife for Music Understanding"
_ICLR.cc/2025/Conference — ICLR 2025 Conference Withdrawn Submission_

### Official Review · Reviewer_yhpD · 2024-10-30

**Soundness:** 1
**Presentation:** 2
**Contribution:** 1
**Rating:** 1
**Confidence:** 5

**Summary:**

This paper trains a Multimodal Large Language Model, OpenMU that is capable of comprehensive music information retrieval tasks. Meanwhile, this paper also proposes OpenMU-Bench, the large-scale dataset and benchmark used to train and evaluate OpenMU. Then, it discusses the design choices of OpenMU and presents experimental results of different hyperparameter settings. Last, this paper compares OpenMU and MU-LLAMA on OpenMU-Bench and shows that OpenMU outperforms MU-LLAMA.

**Strengths:**

This paper proposes a multimodal large language model, OpenMU, capable of comprehensive MIR tasks and outperforms the existing MU-LLAMA model. In addition, this paper establishes the publicly available dataset OpenMU-Bench, which is large-scale and comprehensive.

**Weaknesses:**

The primary weakness of this paper is that its content and experimental results do not convincingly support its claimed contributions. Here are the specific issues:

1. Ambiguity in Contribution between Benchmark and Model: The relationship between the benchmark (including the dataset) and the model is unclear. In the abstract, the benchmark appears to be the main contribution, with the model serving to demonstrate the dataset’s capabilities and provide an example usage. However, in the introduction, this emphasis is reversed: “We aim to contribute to the MIR field by training an MLLM, dubbed OpenMU, for understanding music clips.” and “To address this issue [of data scarcity in the music modality], we construct OpenMU-Bench, a large-scale benchmark for training and evaluating MLLMs in music understanding.”

2. Inadequate Definition of OpenMU-Bench as a Standalone Benchmark: OpenMU-Bench, presented as an evaluation benchmark for MLLMs’ music understanding capabilities, lacks clear task delineation aligned with music understanding competencies. Instead of categorizing tasks by music understanding skill (e.g., captioning, reasoning, genre recognition), they are split by question format, which feels superficial. I suggest structuring tasks more like the MMLU dataset [1], which categorizes tasks by the skills they assess. For instance, genre recognition questions could appear in multiple formats, and ideally, a single, unified metric should represent genre recognition performance.

3. Failure to Provide Model Ranking Capability: A benchmark should allow for a unified score that ranks model performance across music understanding capabilities. Here, OpenMU-Bench lacks a mechanism to yield an overall ranking metric. Tables 3, 4, and 6 present multiple metrics per dataset but fail to combine these into an aggregated score for model ranking.

4. Ill-Defined Tool-Using Task: The “tool-using” task, which the paper claims as a novel contribution, is not well-defined within the benchmark. There is overlap in the music-understanding capabilities it assesses relative to other tasks, and it does not clearly evaluate music understanding itself. A model could perform well on this task merely by calling an API correctly, without demonstrating music comprehension or multi-modal capability. This raises questions: does a high score here reflect the model's music understanding, or simply the quality of the MIR tool it calls?

5. Lack of Comparative Evaluation with Existing Models: The paper explores design choices and hyperparameter variations of its model. However, if the authors aim to present a superior MLLM approach, they should compare its performance against an existing baseline model trained on the same dataset.

[1] Hendrycks, D., Burns, C., Basart, S., Zou, A., Mazeika, M., Song, D., & Steinhardt, J. (2020). Measuring massive multitask language understanding. arXiv preprint arXiv:2009.03300.

**Questions:**

See weakness.

---

### Official Review · Reviewer_C8Cj · 2024-11-01

**Soundness:** 2
**Presentation:** 3
**Contribution:** 2
**Rating:** 5
**Confidence:** 4

**Summary:**

The paper introduces OpenMU-Bench, a comprehensive benchmark suite designed to address the challenge of data scarcity in training multimodal language models for music understanding. It also presents OpenMU, a music understanding model that leverages this benchmark for training. The authors have constructed OpenMU-Bench by creating new annotations and utilizing existing datasets, encompassing a wide range of music understanding tasks such as music captioning, reasoning, lyrics understanding, and music tool usage. OpenMU is shown to outperform baseline models like MU-LLaMA across various tasks, demonstrating its effectiveness in music understanding. Both OpenMU and OpenMU-Bench are open-sourced to foster future research and enhance creative music production efficiency.

**Strengths:**

The paper's strengths lie in its novel contribution to the field of music information retrieval (MIR) through the creation of OpenMU-Bench, a large-scale benchmark that significantly expands the scope of music understanding tasks. The benchmark's comprehensiveness is a notable advantage, as it covers various aspects of music understanding, which is crucial for developing well-rounded multimodal language models. Additionally, the paper demonstrates OpenMU's superior performance over existing models, which is a testament to the effectiveness of the approach taken. The open-sourcing of both the benchmark and the model is another strength, as it promotes transparency, reproducibility, and further development within the research community.

**Weaknesses:**

1. LLark has published its source code at https://github.com/spotify-research/llark, contrary to the paper's claim that it has not open-sourced its models and datasets.

2. OpenMU lacks innovation, being derived from previous works with limited novelty in training.

3. OpenMU-Bench lacks discussion on its construction, including data handling and annotation, limiting its practical application value.

4. The paper does not clarify the overlap between training and testing sets in OpenMU-Bench or explain data selection for training and testing.

5. The paper lacks analysis on why OpenMU outperforms comparative work in some metrics.

6. The paper does not analyze or explain how to avoid bias introduced by processing audio understanding model vectors through Llama.

7. The paper's discussion on evaluation metrics is brief and derived from previous works.

**Questions:**

1. This work lacks comprehensiveness in its evaluation. The paper mentions that LLark's work is similar to what is done here, but LLark has not open-sourced its models and datasets. However, in reality, LLark has published its source code at https://github.com/spotify-research/llark. This work has collected approximately 1 million data points in OpenMU-Bench, which can be reproduced based on the source code, and like OpenMU, a subset of OpenMU-Bench can be used for training to compare the performance of the two models.

2. The core of the proposed model, OpenMU, is derived from the assembly of previous works, and the novelty in the training process is also limited, overall lacking in innovation.

3. OpenMU-Bench, as a primary contribution of this paper, does not have a sufficient discussion on its construction process, including data handling and annotation, and its practical application value is limited.

4. The paper mentions that the model was trained on a subset of the OpenMU-Bench dataset and also uses OpenMU-Bench for evaluation. It does not clarify whether there is an overlap between the training and testing sets, nor does it explain how the data was selected, which data was used for training, and which for testing.

5. Some metrics of OpenMU significantly outperform the comparative work. The paper lacks an analysis of why OpenMU performs so well in these metrics while the comparative work performs poorly.

6. The paper processes the vectors output by the audio understanding model through Llama to transform them into text for analysis and scoring, which inherently introduces bias compared to directly using the vectors from the audio understanding model for distance scoring. However, the paper does not analyze this bias, nor does it explain how to avoid such bias.

7. Moreover, evaluation metrics are an essential component of a benchmark, yet the paper's discussion on the benchmark's evaluation metrics is brief and derived from previous works.

---

### Official Review · Reviewer_XaWv · 2024-11-02

**Soundness:** 3
**Presentation:** 2
**Contribution:** 1
**Rating:** 3
**Confidence:** 4

**Summary:**

The paper presents OpenMU and OpenMU-Bench. OpenMU-Bench, constructed by leveraging existing datasets and generating new annotations with GPT-3.5, is a large-scale benchmark suite for music understanding tasks like captioning, reasoning, lyrics understanding, tool using, and multiple-choice questions. OpenMU is a music understanding model trained with specific procedures. It outperforms baseline models like MU-LLAMA. Both OpenMU and OpenMU-Bench are open-sourced to contribute to music understanding research and improve creative music production.

**Strengths:**

1. Standardizing the evaluation metrics for text-generation tasks in OpenMU-Bench is a smart step. It enhances the consistency and fairness of benchmarking, allowing for more effective comparisons between various music-understanding models.
2. The thorough exploration of key factors in training OpenMU is useful. For instance, examining how the number of music tokens impacts training efficiency and model convergence provides reusable insights for future research.

**Weaknesses:**

1. The proposed model and dataset primarily rely on established techniques and common practices in the field, resulting in a lack of novelty:
1) Although the paper utilizes existing datasets and employs GPT-3.5 to generate new annotations, the underlying data sources are mainly based on pre-existing music-related datasets, with no introduction of new music data or innovative data-collection methods.
2) Regarding the model architecture, the use of AudioMAE for encoding music clips, Llama 3 as the language model, and a specific music-language projector all involve existing techniques.
2. In terms of employing LoRA adapters and the approach of using APIs to address the limitations of large language models, these are both strategies found in other research on multimodal language models.
3. There are issues with the presentation. For instance, the section discussing the number of music tokens and LoRA-related content (Section 5.2) is included in Section 4 without sufficient elaboration, which may confuse readers when they first encounter the training details.
4. To improve clarity, it would be helpful to distinguish music as a field rather than a modality. In the paper, music is referred to as a modality. However, music is a field that includes both symbolic and audio data. This paper focuses solely on audio, which is the true modality under discussion.

**Questions:**

1.Is there a quantitative indicator or evaluation framework specifically designed to measure the consistency of GPT-3.5-generated text data with metadata across different datasets? If so, could you elaborate on its design principles and the key considerations in the evaluation process?

2. MERT is pretrained on 160K hours of music data. Given this, why is AudioMAE, which is pretrained on only 3K hours of music data (as noted in the paper), being used?

---

### Official Review · Reviewer_enpB · 2024-11-04

**Soundness:** 3
**Presentation:** 4
**Contribution:** 2
**Rating:** 5
**Confidence:** 5

**Summary:**

This paper presents a benchmark for evaluating multimodal language models in music understanding, along with a model trained on this benchmark data collection. The authors introduce a data collection derived from existing resources with additional annotations; the key topics covered by the benchmark; and a Llama3-based model trained on this collection. Additionally, the paper provides an initial evaluation to compare the performance of the proposed model with other baseline models.

**Strengths:**

This paper has two strengths:

1.  The authors make significant efforts in collecting and annotating music datasets using LLM. This contribution provides valuable resources for the music community, supporting further development of LM-based music understanding models.

2.  The evaluation of the proposed OpenMU is thorough, with comparisons to baseline models such as Mu-Llama, MusiLingo, and M2UGen. The paper also includes ablation studies examining the effects of token size/length and various low-rank adaptation parameters/layers on OpenMU's performance.

**Weaknesses:**

The paper contains two weaknesses:

1. **Limited Novelty**: The contributions of this paper are somewhat constrained. It should focus on either advancing the benchmark for music understanding or improving LM-based music understanding models. While the authors have invested considerable effort in data collection, task formulation, and initial evaluation of the benchmark, there are no new designs for model architecture or evaluation metrics except a new task termed "tool using". Furthermore, it should be noted that many mentioned topics or designs in this paper is not original; the LM-based music understanding model can be traced back to prior works [1][2], and the music understanding benchmark can also refer to [3][4]. As a result, this paper appears to combine two areas without offering new insights. It is challenging to assess the paper's contributions, as we cannot ascertain its acceptance of both NLP and MIR researchers without reviewing its benchmark interface (e.g., a website) and its feasibility. Given the focus of the paper, it may be more appropriate to submit it to a conference or workshop related to datasets and benchmarks, such as the NeurIPS dataset and benchmark track, rather than the ICLR regular track.

2. **Insufficient Evaluation Metrics for Pure Music Understanding**: The evaluation metrics presented in the paper primarily address the music captioning task from a natural language processing perspective (e.g., BLEU score, ROUGE score, and METEOR score). In the field of music information retrieval (MIR), there is a need for more specialized metrics that address specific tasks such as tempo estimation, chord recognition, melody extraction, and genre classification. While many of these metrics could be evaluated within OpenMU-Bench, this paper does not provide them as previous works have done [3][4]. The advantage of a multimodal language model (or music language model) over traditional MIR models lies in its ability to seamlessly integrate knowledge from both NLP and music. However, the current OpenMU-Bench falls short of this standard.

In summary, while I appreciate the effort put into this paper, I remain confidence on its contributions to the dataset and benchmark areas but skeptical about its suitability for the ICLR conference.

[1] LLark: A Multimodal Instruction-Following Language Model for Music

[2] MusiLingo: Bridging Music and Text with Pre-trained Language Models for Music Captioning and Query Response

[3] The Song Describer Dataset: a Corpus of Audio Captions for Music-and-Language Evaluation

[4] MARBLE: Music Audio Representation Benchmark for Universal Evaluation

**Questions:**

1. Regarding the results presented in Table 5, were all the other models (MusiLingo, Mu-Llama, M2UGen) trained specifically on OpenMU-Bench? Or just their pretrained models from their respective sources and were directly evaluated on OpenMU-Bench?

2. What is the licensing status of the data collections in OpenMU-Bench? Are they restricted to research-only usage, or are they also available for commercial purposes? And are these licenses from different datasets consistent?

3. Do you have any analyses or insights regarding the hallucination tendencies of LLMs when generating annotations? Specifically, have you observed any incorrect annotations produced by GPT-3.5, and do you employ any filtering methods to ensure the quality of the dataset?

4. Is there a possibility of creating an anonymous website or repository for accessing the benchmark? This would enhance concerns regarding its practical usability.

---

### Note · Authors · 2024-11-12

I have read and agree with the venue's withdrawal policy on behalf of myself and my co-authors.